# Association between Body Fat and Elevated Blood Pressure among Children and Adolescents Aged 7–17 Years: Using Dual-Energy X-ray Absorptiometry (DEXA) and Bioelectrical Impedance Analysis (BIA) from a Cross-Sectional Study in China

**DOI:** 10.3390/ijerph18179254

**Published:** 2021-09-02

**Authors:** Manman Chen, Jieyu Liu, Ying Ma, Yanhui Li, Di Gao, Li Chen, Tao Ma, Yanhui Dong, Jun Ma

**Affiliations:** Institute of Child and Adolescent Health, School of Public Health, Peking University, Beijing 100191, China; 1911210173@pku.edu.cn (M.C.); jieyulynne@163.com (J.L.); mypku232@163.com (Y.M.); yanhui_lyh@163.com (Y.L.); gaodi1993@163.com (D.G.); clcl@bjmu.edu.cn (L.C.); 1610306216@pku.edu.cn (T.M.)

**Keywords:** body fat, elevated blood pressure, dual-energy X-ray absorptiometry, bioelectrical impedance analysis, childhood

## Abstract

To investigate the associations between body fat percentage (BF%) with childhood blood pressure (BP) levels and elevated BP (EBP) risks, and further examine the validity of bioelectrical impedance analysis (BIA), we conducted a cross-sectional study of 1426 children and adolescents aged 7–17 years in Beijing, 2020. EBP, including elevated systolic BP (ESBP) and elevated diastolic BP (EDBP), was defined based on the age- and sex-specific 90th BP reference values of children and adolescents in China. BF% was measured by dual-energy X-ray absorptiometry (DEXA) and BIA devices, and was divided into four quartiles. Log-binomial models were applied to calculate odds ratios (ORs) and 95% confidence intervals (95%CI). Girls tended to have higher BF% levels than boys (*p* < 0.05). There was 41.0% of girls who developed EBP. High BF% was associated with increased BP levels with ORs of 0.364 (95%CI = 0.283–0.444) for SBP, 0.112 (95%CI = 0.059–0.165) for DBP, and 1.043 (95%CI = 1.027–1.059) for EBP, while the effects were more pronounced in girls and older-aged children. BIA devices agreed well with BF% assessment obtained by DEXA. High BF% might have negative effects on childhood BP. Convenient measurements of body fat might help to assess childhood obesity and potential risks of hypertension.

## 1. Introduction

The global prevalence of hypertension or elevated blood pressure (EBP) in children was on a long-term upward trend, seriously endangering health [1]. Blood pressure (BP) in childhood will physiologically increase with age, and will still not be basically stable until adulthood. Previous studies had found that the mendelian inheritance can be confirmed in approximately 20% of families [2] and 60% of twin studies [3] in familial hypertension cases. Hypertension has caused a serious disease burden in China, and is associated with adverse cardiac changes, adipose tissue factor changes, and vascular damage [4]. The increase in BP is affected by multiple factors, such as excessive intake of salt, reduced physical activity, and excessive psychological stress, which could further account for the development of hypertension [5]. Overweight and obesity has been recognized as the most important risk factor for hypertension. People who are overweight and obese tend to accumulate too much fat and to be at risk of hypertension [6].

The current research on body fat and BP in children and adolescents in China mainly uses body mass index (BMI) as the evaluation standard, and rarely involves diversified body fat content components. However, a single BMI index cannot well-predict body fat distribution and the risk of related metabolic diseases [7], such as hypertension. Therefore, the health problems caused by body fat distribution should be paid more attention to. High body fat percentage (BF%) has been shown to be a cardiovascular risk factor among children and adults’ populations [8]. A cohort study including medical check-up examinees aged 20 years or older found that high BF% (76–100th percentile) was related with a high incidence of hypertension, compared to low BF% (1–25th percentile) [9]. In addition, in a study with inclusion of children and adolescents aged 6 to 16 years, the control of body fat mass index (FMI) or fat mass percentage (FMP) below P70 contributed to the decline of BP levels and blood glucose metabolism abnormalities by 8.53% to 43.24% [10]. Allowing for the physiological increase of children’s BP and the prevalence of hypertension with the increase of obesity, it is rather crucial to establish the associations between the actual body fat composition and BP for the guidance of hypertension prevention and reasonable control of body fat in children.

At present, the techniques most commonly applied to assess body composition are underwater weighing, dual-energy X-ray absorptiometry (DEXA), bioelectrical impedance analysis (BIA), and computer tomography (CT). The DEXA was considered to be the gold standard for body composition measurement, since it can measure body composition more accurately with advantages of stability and repeatability [11,12]. However, limitations do exist as the DEXA is costly, non-portable, and often necessitates training by a licensed technician due to the existence of certain radioactivity. BIA has evolved to include the use of multiple frequencies and impedance measurements to improve the accuracy and reliability of body composition estimates [13]. BIA is relatively simple, portable, and noninvasive [14,15]. Unlike underwater weighing or DEXA, BIA does not require total body submersion in water or exposure to radiation. In addition, multifrequency BIA machines (e.g., InBody770) have been developed for assessing segmental and total body composition [16]. Therefore, it is necessary to investigate the reliability and validity of different BIA devices before initiation of field studies on pediatric body composition.

This cross-sectional study of Chinese children and adolescents (age 7–17 years) aims to examine the accuracy and validity of two portable BIA devices by comparing their results with those from DEXA measurements, and further estimate the association between BF% with BP levels and EBP risks across gender and age.

## 2. Materials and Methods

### 2.1. Study Population

Data were obtained from a cross-sectional study conducted in children and adolescents from Beijing of China in 2020. The design adopted a stratified cluster random sampling method, and selected all the participants from four schools, including elementary school, junior high school, and high school. A pre-survey was conducted first and a total of 1426 children and adolescents aged 7–17 years were invited to participate in the project. For the inclusion and exclusion criteria, the school doctor introduced the research purpose and content of the project to all the children and adolescents and their parents in detail, and the investigation was carried out only after the students and their parents signed the informed consent. The research of this project has been reviewed and approved by the Ethics Committee of Peking University (Number: IRB00001052-20024).

### 2.2. Data Collection and Measurements

The research content included general body measurements and a simple questionnaire survey. General body measurements included measurements of height, weight, waist circumference, blood pressure, and body composition, which were carried out by professionally trained medical examiners. Before conducting the survey, the spreads were carried out, and all the questionnaire questions were strictly checked. The questionnaire was distributed by the project team through the school doctors to the parents of the students before the physical examination (or through an online questionnaire), with the exception of children at third grade or under primary school, who completed the questionnaires at home with their primary guardian. The teacher collected the completed questionnaire on the day of the physical examination and handed it over to the project team members. Child-reported questionnaires collected included birth date, gender, vegetable and fruits consumption, smoking habits, alcohol drinking, sleeping duration, and mid–high sports hours. The age was calculated as (date of examination—date of birth)/365.25, and we divided the age into three groups (7–12 years, 13–15 years, and 16–17 years). Generally, these three age groups represented the primary school stage (grade 1 to 6), middle school stage (grade 1 to 3), and middle high school stage (grade 1 to 3) in China.

Height was measured by using a uniform and calibrated mechanical stadiometer (model TZG, Jiangyin No. 2 Medical Equipment Factory, Jiangsu, China), with an accuracy of 0.1 cm. At the same time, the participants were standing straight and barefoot. Weight was measured by using a uniform and calibrated electronic scale (model RGT-140, Shanghai Dachuan Electronic Weighing Apparatus Co. Ltd., Shanghai, China) to the nearest 0.1 kg while subjects were wearing short clothes and standing naturally in the center of the weight measuring plate to keep the body stable. According to the instructions for use, we checked its working status, accuracy, and sensitivity before use. Waist, bust, and hip circumferences were measured using myotape (MT05 by AccuFitness, Green Village, Colorado, USA) to the nearest 0.1 cm, taking measurements in pairs. Blood pressure was measured using a unified medical electronic sphygmomanometer certified by the national standard scheme, mercury sphygmomanometers (model XJ11D, Shanghai Medical Instruments Co. Ltd., Shanghai, China), and stethoscopes (model TZ-1, Shanghai Medical Instruments Co. Ltd., Shanghai, China). Participants were asked to measure the right arm and sit quietly for at least 5 min before the first reading. Systolic blood pressure (SBP) was determined by onset of the first Korotkoff sound and diastolic blood pressure (DBP) was determined by the fifth Korotkoff sound.

Body composition was measured by professional medical personnel using a GE Healthcare Lunar iDXA dual-energy X-ray bone densitometer in accordance with the standard use process and program requirements described by the instrument, scanning the whole body and collecting images. The participants were placed as required, lying flat on the scanning bed, with the body in the middle of the instrument, with the thumb facing up, and the palm facing but not touching the leg. In addition, all BIA measurements were performed by trained research assistants, where six-frequency eight-electrodes (InBody770, Biospace, Seoul, Korea) and dual-frequency eight-electrodes (Huawei scale 3 Pro, Huawei, Shenzhen, China) BIA machines were adopted to measure children’s body composition.

All measurements were logically checked before examination. During each on-site physical examination, a special person was assigned to conduct on-site supervision to ensure that the measurement methods and records of each measurement index were correct and standardized.

### 2.3. Definitions of EBP, Body Fat Percentage, and Adjusted Factors

The averages of SBP and DBP values were calculated by two measurements. According to the national reference of EBP in China [17], ESBP (elevated systolic blood pressure) and EDBP (elevated diastolic blood pressure) were defined as average measured SBP and DBP greater than or equal to the 90th percentile (on the basis of age, sex, and height percentiles), respectively. EBP was defined as ESBP or EDBP of children and adolescents. BF% was divided into quartiles based on the *P*_25_, *P*_50_, and *P*_75_ cutoff value among participants of this study. According to the quartile method, we divided the BF% into Q1 (1–25th percentile), Q2 (26–50th percentile), Q3 (51–75th percentile), and Q4 (76–100th percentile).

Some confounding factors were also included in this study, such as cigarette smoking (yes/no), alcohol consumption (yes/no), and so on. We also adjusted for some eating habits, sleeping time, and exercise time. The vegetables and fruits intake were calculated as: daily intake = (frequency (days per week) × daily servings)/7. The daily middle and high physical activity time = frequency (days per week) × duration (hours per time). Daily sleeping time was based on the Pittsburgh Sleep Scale.

### 2.4. Statistical Analysis

All analyses were performed with R version 3.5.1 (R Foundation for Statistical Computing, Vienna, Austria). Continuous variables were expressed by mean values and standard deviations, and categorical variables were expressed by numbers and percentages. We used a scatter plot to intuitively estimate the correlation between BF% and BP level among boys and girls. Bland–Altman plots were created using mean and the difference to predict the limits of agreement of body fat mass and BF% between DEXA and dual-frequency BIA machine measurements. Multivariate linear models were applied to calculate incident rates and 95% confidence intervals (95% CI) for BP level and risk of EBP. We also assessed the prevalence of EBP, ESBP, and EDBP and their associations (odds ratios (ORs)) with BF% by Log-binomial models. All models were adjusted for age, gender, vegetables and fruits intake, whether smoking and drinking or not, sleeping, and middle and high physical activity time. Furthermore, gender- and age-stratified analyses were conducted to evaluate the potential differences. We considered the associations to be statistically significant when the two-sided *p*-value was less than 0.05.

### 2.5. Sensitivity Analysis

To detect the similarity or difference between BF% measured by DEXA and two BIA devices, and to evaluate its associations with BP level and EBP risks, we performed sensitivity analyses based on the same samples and regression models. Pearson’s correlation was used to assess the correlations between body fat mass measured, while controlling for potential confounders. Overall, good correlation between body fat mass obtained by DEXA and assessed by the six-frequency BIA machine (InBody770) was observed (r = 0.982, *p* < 0.001), and the correlation between body fat mass obtained by DEXA and estimated by the dual-frequency BIA machine (Huawei scale 3 Pro) was r = 0.973, *p* < 0.001. Since it is convenient for children and adolescents at home, for simplicity, this study selected body composition measured by the dual-frequency BIA machine (Huawei scale 3 Pro) to estimate the associations between BF% with BP level and EBP risks for the sensitivity analysis.

## 3. Results

At the baseline, the mean age of the 1426 participants was 11.96 ± 3.10 years and mean BMI was 20.44 ± 4.70 kg/m^2^, and the mean body fat percentage was 29.36% ± 8.22% with the median of 29.60 (23.80–35.40). There were 717 boys (50.30%) and 709 girls (49.70%) (Table 1). Using the method of DEXA, boys tended to have lower body fat mass (14.45 ± 8.63) than girls (15.85 ± 7.85), and similar trends were found with the two BIA devices (six-frequency and dual-frequency). Unsurprisingly, gender differences of BF% were also observed, and girls were prone to have higher BF% than boys, whether it was measured by DEXA or BIA devices (all *p* ≤ 0.001). Boys tended to have higher SBP (sitting position) compared with girls (*p* = 0.001), while no significant differences were observed for DBP (sitting position) between boys and girls (*p* = 0.269). In addition, boys had a high risk of EBP (*p* = 0.001) and high ESBP (*p* < 0.001), while girls were prone to have high EDBP (*p* = 0.001).

Figure 1 showed multivariable relationships of SBP and DBP with body fat percentage among boys and girls. It was clear that the associations were pronounced among the total population (SBP: r = 0.089, *p* < 0.001; DBP: r = 0.089, *p* < 0.001), but the effects were more significant when we restricted to girls (SBP: r = 0.309, *p* < 0.001; DBP: r = 0.185, *p* < 0.001), rather than boys (SBP: r = 0.035, *p* = 0.345; DBP: r = −0.01, *p* = 0.776).

As shown in Figure 2, based on Bland–Altman plots, although within the limits of agreement (mean ± 1.96 (SD)), the values less than a mean of 2.03% body fat mass were over-predicted and the values above 2.73% were under-predicted with the DEXA and dual-frequency BIA machine measurements. A similar trend was observed for BF%, where most of the participants were within the limits of agreement (2.24% BF% were over-predicted, 1.40% were under-predicted).

The association between BF% and the incidence of BP level across gender and age is shown in Table 2 and Table 3. As continuous variables, high body fat percentage was associated with increased SBP (*β* = 0.364, 95% CI = 0.283–0.444) or DBP (*β* = 0.112, 95% CI = 0.059–0.165) levels, the effects were more pronounced in girls and those with older ages. We next divided all subjects into four groups by the quartiles of BP levels and estimated the associations by using the corresponding lowest quartile as a reference. Graded associations between BP categories and high BF% were found. In particular, significantly increased risks of high BF% were observed for the highest quartile of SBP both for different gender (*β* = 8.094, 95% CI = 6.243–9.945) and those older-aged children (*β* = 12.376, 95% CI = 6.764–17.988). Similar associations detected by the dual-frequency BIA device are described in Appendix A.

The ORs and 95% CI of BF% and the risks of EBP, ESBP, and EDBP by gender and age are shown in Figure 3 and Figure 4. In both boys and girls, high BF% was significantly related with higher ORs (OR > 1, *p* < 0.05) for EBP and ESBP compared with the normal BP group, whether considering body fat percentage as a continuous or categorical variable. As for different age groups, the ORs of body fat percentage for EBP and ESBP among 7–12-year-old participants were relatively low, and among 16–17-year-old participants reached the highest level, with the OR values of 1.121 (1.051, 1.196) and 1.137 (1.063, 1.216), respectively. Additionally, high BF% measured by the dual-frequency BIA device was positively related with EBP, ESBP, and EDBP (Appendix A).

## 4. Discussion

In this study, girls tended to have higher BF% and body fat mass than their boy counterparts. Increased BF% was positively associated with BP levels, EBP, and ESBP risks, after adjusting for various potential confounding factors, and these effects were more significant in girls and older-aged children. Besides, BIA devices agreed well with the BF% assessment obtained by DEXA.

In the present study, the risk of having EBP increased with increasing levels of BF%, but ESBP and body weight shared the highest correlation. These results were similar with the results of earlier studies, in which excess BF% was used to examine the relationship between BP and adiposity in children and adolescents [18,19]. One study conducted that increasing BF% was significantly associated with the increased risk of hypertension, even in non-obese individuals [20]. Particularly, previous studies found that high SBP was frequently related to higher adiposity in children [21,22]. One study showed several anthropometric variables (including stature, weight, BMI, waist circumference (WC), triceps, subscapular, gluteal skinfold thicknesses, and percentage body fat) that were significantly correlated with SBP in the children in Eastern Cape province, South Africa [23]. In a group of Barbadian children, BMI z scores were also independently correlated with SBP across both boys and girls [24]. Furthermore, further studies are warranted to determine the most vulnerable BP index that is associated with high BF%.

Some underlying mechanisms could account for the fact that the associations of the body composition measurements with BP were inconsistent for SBP and DBP in the stages of childhood and adolescence. Previous studies suggested that the root cause of high SBP in subjects with obesity was primarily due to a combination of factors that raise systemic vascular resistance [25]. The coordinated appearance of insulin resistance and high BP in obese individuals led several authors to hypothesize that insulin resistance is one of the major determinants of increased systemic vascular resistance in obesity [25]. In support of the theory above, Sinaiko found a relationship between fasting insulin and SBP among children and adolescents [26]. Insulin could cause hypertension through stimulation of the sympathetic nervous system, an increase in renal sodium retention, modulation of calcium transport, and consequent induction of hypertrophy of vascular smooth muscle [27]. However, DBP is determined by the elasticity and resistance of arteries. Aging lowers DBP because arterial elasticity decreases. Exposure to obesity early in life may induce changes in the arteries, contributing to the development of atherosclerosis in adulthood [28]. A recent study suggested that increased intima media thickening occurred more in obese children [29], and probably produces decreased vascular elasticity in adolescents which then tracks into adulthood [30]. These findings may partially explain why adiposity measures are not related to DBP in adolescents in our study, and indicate that higher BF% may represent an insulin resistance to raise systemic vascular resistance, and contribute to EBP and especially ESBP in children and adolescents.

There are anthropometric, metabolic, and cardiovascular differences between boys and girls prior to the appearance of any external signs of puberty. The fact that the girls had higher BMI levels than boys is consistent with expectations since women possess more sex-specific fat than men at virtually all ages [31,32]. The mechanisms of the observed gender differences are not clear at present, though as expected, girls classified themselves as less active compared to boys [33]. Ayyavoo and colleagues indicated that girls had higher adrenal androgen concentrations and greater adiposity than boys throughout childhood, which accounted for the differences in insulin sensitivity [34]. Apart from this, increased fat mass was also likely associated with higher nocturnal BP in girls [34]. Differences in lifestyles (including diet habits, outdoor activities, etc.) may also account for the sex difference in the associations of BF% with BP, so the potential sex-specific associations need to be elucidated further. Another significant finding for the present study is that the relationship between BF% and BP was more significant in the older children than in the younger children. This result is consistent with the literature and suggests that older children had higher values of body composition [35]. More importantly, previous studies indicated that the associations between central fat and BP increase as children mature [36]. Therefore, the current study indicated that more attention should be paid to girls and older children, who may be more vulnerable to high BF%.

The results of our preliminary study indicated the reliability of six-frequency or dual-frequency BIA machines, and these could be used to mirror BF%. They were non-invasive, inexpensive, and applicable measurements using these BIA devices, that were superior methods for the estimation of total body composition. A previous study suggested that multi-frequency devices could lead to better precision than single-frequency, because they might be less subject to error caused by redistribution of total body water between extracellular water and intracellular water [37]. This statement was further supported by research that has shown that multi-frequency BIA devices have better agreement with reference techniques such as DEXA than single-frequency devices among school-aged children [38]. Therefore, the application of a BIA method for mirroring childhood BF% may be a viable alternative to the DEXA, especially for similar kinds of school-based surveillance. However, prior studies using BIA devices have demonstrated suitable estimates of body composition, despite underestimating fat mass and overestimating fat-free mass in adults when compared to DEXA [39,40]. Therefore, further studies are necessary to confirm our findings in different ethnic groups from other parts of the world.

The main strengths of our study include the usage of DEXA, a previously validated measure of BF% in pediatric populations [41]. Additionally, we compared BIA machines with DEXA measurements, and proposed the accuracy of multi-frequency BIA devices. However, certain limitations should be considered. The sample size was not large enough, which limited the extrapolation of the conclusion. Additionally, when assessing adiposity during childhood and adolescence, even considering age adjustments, different pubertal stages and hormonal changes should also be considered as a determinant or confounding variable, and therefore the absence of this data could be considered in the following study. When measuring with DEXA, the patient is exposed to radiation, and this method is expensive and time-consuming. An alternative simple method such as BIA measurement appears to be a useful and feasible tool, only if it is used in the same ethnic group, same age, with the same device [42,43]. In addition, it was stated that BIA devices may prevent the incorrect diagnosis of obesity as determined by BMI alone, especially during the pubertal period [44].

This observation has important clinical and public health implications. Excessive body fat is detrimental to body health and physical performance. A higher body fat percentage indicated a higher level of cardiovascular risks, such as hypertension demonstrated in our study, dyslipidemia, and type 2 diabetes [45,46,47]. As obesity has become a serious public health problem affecting children, it is important to accurately monitor the body composition of children in order to safeguard against future health diseases. These findings underscore the importance of accurately examining childhood obesity in relation to the functional outcomes. Thus, the high BF% observed in this study has implications for the early prevention of cardiovascular diseases. Intensive and precise intervention programs should be instituted in schools to prevent and control possible excessive body fat among school-aged children and adolescents, and more attention should be paid to girls and older-aged children, who might be more vulnerable to high BF%.

## 5. Conclusions

We underscore the importance of accurately examining childhood obesity in relation to the functional outcomes. Thus, the high BF% observed among the children in this study has implications for the early prevention of cardiovascular diseases. Intensive and precise public health programs should be instituted in schools to monitor the body composition among school-aged children and adolescents, and more attention should be especially paid to girls and older children. In addition, BIA devices agreed well with the BF% assessment obtained by DEXA. Convenient measurements of body fat might help to acutely assess childhood obesity and potential risks of hypertension and related early cardiovascular diseases.

## Figures and Tables

**Figure 1 ijerph-18-09254-f001:**
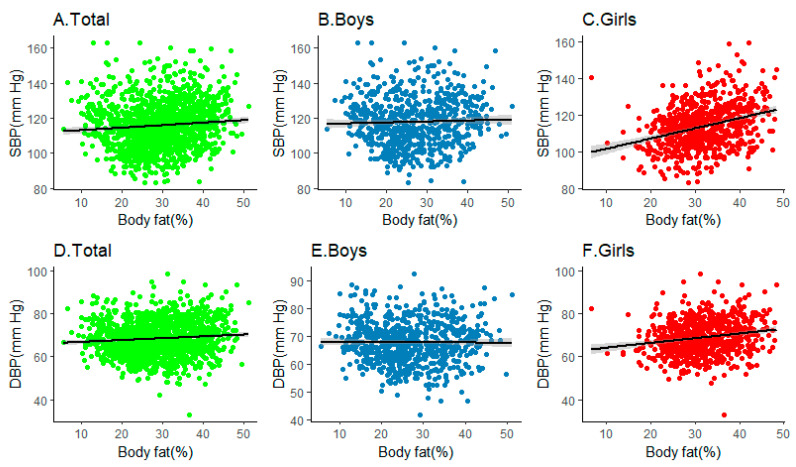
Multivariable relationships between SBP or DBP with BF%among children and adolescents. Note: (**A**–**C**) represent the relationship between SBP with BF% in total, boys and girls; (**D**–**F**) represent the relation-ship between DBP with BF% in total, boys and girls.

**Figure 2 ijerph-18-09254-f002:**
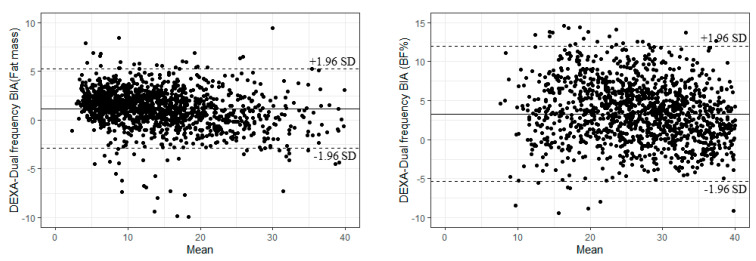
Comparison of body fat mass and BF% measured using DEXA and the dual-frequency BIA machine, Bland–Altman plots.

**Figure 3 ijerph-18-09254-f003:**
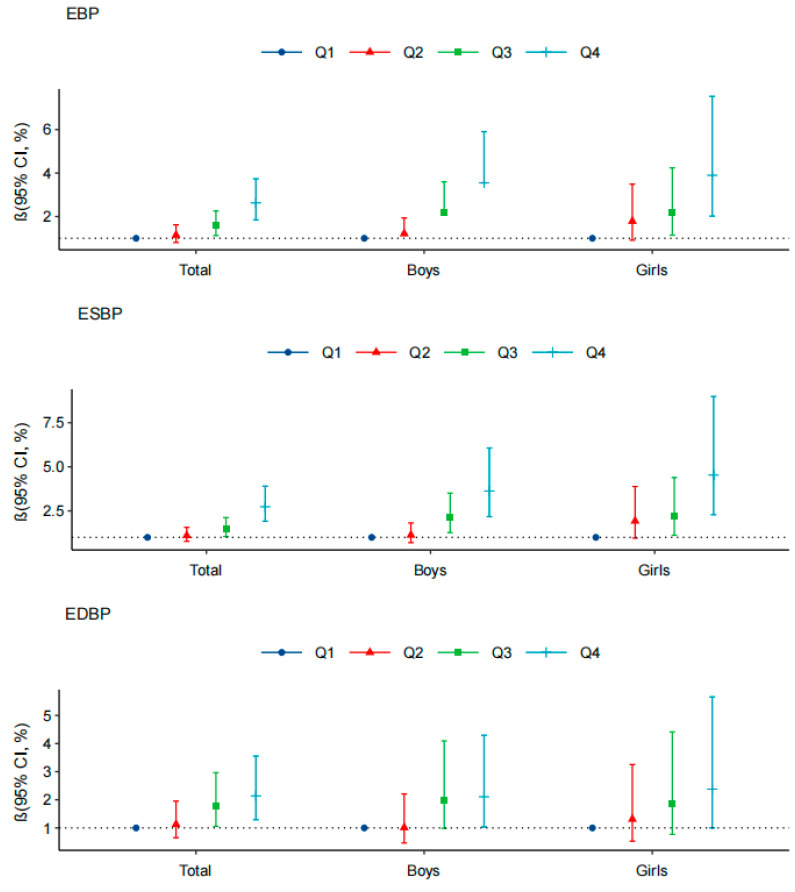
Association between body fat percentage (%) and risk of EBP, ESBP, and EDBP among boys and girls. Note: Model was adjusted for age, gender, vegetable consumption, fruit consumption, smoking habits, alcohol drinking, sleeping duration, and mid–high sports hours.

**Figure 4 ijerph-18-09254-f004:**
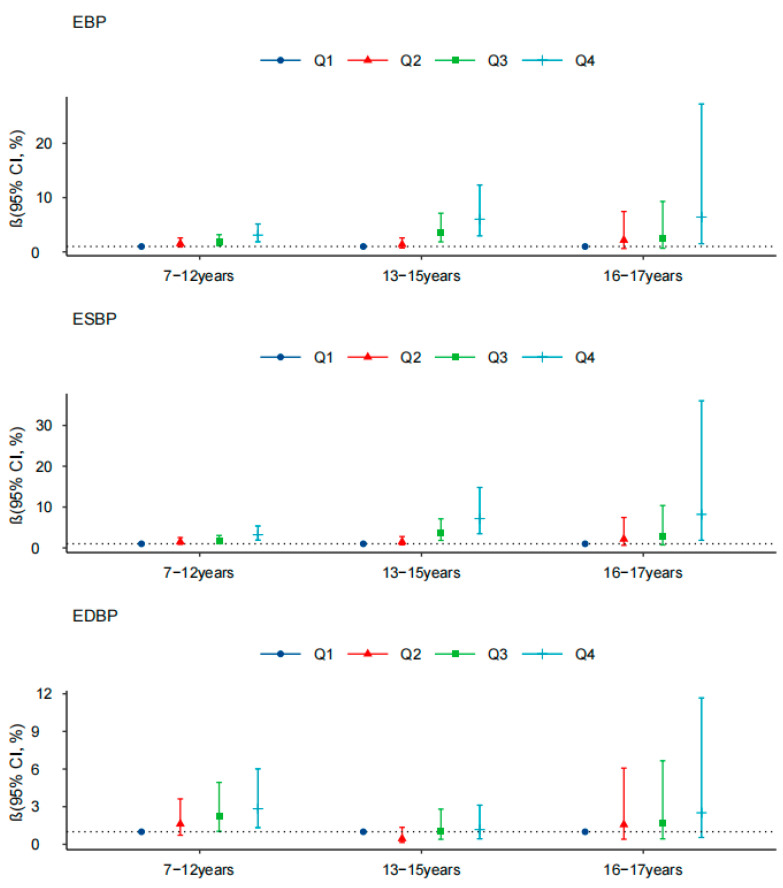
Association between body fat percentage (%) and risk of EBP, ESBP, and EDBP among different age groups. Note: Model was adjusted for age, gender, vegetable consumption, fruit consumption, smoking habits, alcohol drinking, sleeping duration, and mid–high sports hour.

**Table 1 ijerph-18-09254-t001:** General characteristics of included participants.

	Overall	Boys	Girls	*t* Value	*p-*Value *
Total (*n* (%))	1426	717 (50.3)	709 (49.7)		
**Anthropometrics and age**					
Height (cm)	154.08 ± 15.61	156.57 ± 17.19	151.56 ± 13.38	96.063	<0.001
Weight (kg)	50.07 ± 18.22	52.4 ± 19.84	47.72 ± 16.11	30.359	<0.001
BMI (kg/m^2^)	20.44 ± 4.70	20.64 ± 4.76	20.24 ± 4.64	0.059	0.808
Waist circumference (cm)	70.49 ± 12.02	72.65 ± 12.63	68.29 ± 10.94	10.917	0.001
Chest circumference (cm)	77.54 ± 12.90	77.97 ± 12.80	77.09 ± 13.00	0.040	0.842
Hip circumference (cm)	87.42 ± 24.73	87.10 ± 12.80	87.75 ± 32.63	0.585	0.445
Age (year), mean ± SD	11.96 ± 3.10	11.94 ± 3.12	11.98 ± 3.09	0.024	0.876
7–12 years	745 (52.2)	380 (53.0)	365 (51.5)		
13–15 years	455 (31.9)	220 (30.7)	235 (33.1)	1.035	0.596
16–17 years	226 (15.8)	117 (16.3)	109 (15.4)		
**DEXA**					
Body fat mass (kg)	15.15 ± 8.28	14.45 ± 8.63	15.85 ± 7.85	−3.199	0.001
Median (*P*25, *P*75)	13.49 (8.83, 19.20)	12.08 (8.17, 18.25)	14.48 (9.97, 20.00)		
Body fat percentage (%)	29.36 ± 8.22	26.90 ± 8.83	31.85 ± 6.69	−11.931	<0.001
Median (*P*25, *P*75)	29.60 (23.80, 35.40)	26.00 (20.50, 33.60)	31.50 (27.20, 36.40)		
**Six-frequency BIA machine**					
Body fat mass (kg)	13.39 ± 8.79	12.41 ± 8.82	14.39 ± 8.66		
Median (*P*25, *P*75)	11.40 (6.60, 17.40)	9.90 (6.20, 16.10)	13.10 (7.80, 19.00)	−4.279	<0.001
Body fat percentage (%)	25.26 ± 9.40	22.54 ± 9.33	28.00 ± 8.65	−11.454	<0.001
Median (*P*25, *P*75)	24.70 (17.90, 32.30)	21.20 (15.10, 29.70)	28.20 (21.50, 34.20)		
**Dual-frequency BIA machine**					
Body fat mass (kg)	13.96 ± 8.85	13.03 ± 8.82	14.89 ± 8.78	−3.999	<0.001
Median (*P*25, *P*75)	12.16 (7.28, 18.33)	10.68 (6.56, 16.97)	13.41 (8.14, 19.52)		
Body fat percentage (%)	26.07 ± 9.19	23.46 ± 9.08	28.70 ± 8.54	−11.210	<0.001
Median (*P*25, *P*75)	25.88 (19.28, 32.47)	22.50 (16.55, 29.97)	28.81 (22.97, 35.00)		
**Blood pressure level**					
SBP, sitting position (mmHg)	115.77 ± 13.09	117.76 ± 13.76	113.76 ± 12.05	10.770	0.001
DBP, sitting position (mmHg)	68.33 ± 7.80	67.85 ± 7.59	68.81 ± 7.98	1.221	0.269
BP (*n* (%))					
Normal BP	780 (54.7)	362 (50.5)	418 (59)	10.317	0.001
Elevated BP	646 (45.3)	355 (49.5)	291 (41)		
SBP (*n* (%))					
Normal SBP	810 (56.8)	373 (52)	437 (61.6)	13.428	<0.001
Elevated SBP	616 (43.2)	344 (48)	272 (38.4)		
DBP (*n* (%))					
Normal DBP	1234 (86.5)	643 (89.7)	591 (83.4)	12.230	0.001
Elevated DBP	192 (13.5)	74 (10.3)	118 (16.6)		

BMI: body mass index; DEXA: dual-energy X-ray absorptiometry; BIA: bioelectrical impedance analysis; SBP: systolic blood pressure; DBP: diastolic blood pressure. * *p*-value for comparison between boys’ and girls’ estimates was based on t-tests for continuous variables and chi-square tests for categorical variables.

**Table 2 ijerph-18-09254-t002:** Association between body fat percentage (%) and incidence of blood pressure level among boys and girls.

	Overall	Boys	Girls
**DEXA**			
SBP			
Continuous	0.364 (0.283, 0.444) **	0.420 (0.313, 0.527) **	0.470 (0.342, 0.599) **
Q1	Reference	Reference	Reference
Q2	1.229 (−0.605, 3.063)	1.524 (−0.817, 3.864)	2.725 (−0.443, 5.893)
Q3	3.310 (1.466, 5.154) **	4.934 (2.432, 7.436) **	4.276 (1.177, 7.376) *
Q4	8.094 (6.243, 9.945) **	9.769 (7.260, 12.277) **	9.276 (6.152, 12.399) **
*p*-value for trend	<0.001	<0.001	<0.001
DBP			
Continuous	0.112 (0.059, 0.165) **	0.089 (0.020, 0.157) *	0.166 (0.077, 0.256) **
Q1	Reference	Reference	Reference
Q2	−0.145 (−1.351, 1.060)	0.081 (−1.418, 1.581)	−0.233 (−2.435, 1.970)
Q3	0.349 (−0.863, 1.561)	−0.135 (−1.737, 1.468)	0.835 (−1.320, 2.991)
Q4	2.593 (1.377, 3.810) **	3.019 (1.412, 4.626) **	2.503 (0.331, 4.674) *
*p*-value for trend	<0.001	0.012	<0.001

* *p* < 0.05; ** *p* < 0.001 Model was adjusted for age, gender, vegetable consumption, fruit consumption, smoking habits, alcohol drinking, sleeping duration, and mid–high sports hours.

**Table 3 ijerph-18-09254-t003:** Association between body fat percentage (%) and incidence of blood pressure level among different age groups.

	7–12 years	13–15 years	16–17 years
**DEXA**			
SBP			
Continuous	0.392 (0.279, 0.506) **	0.520 (0.379, 0.662) **	0.688 (0.459, 0.918) **
Q1	Reference	Reference	Reference
Q2	1.836 (−0.617, 4.29)	2.000 (−1.27, 5.269)	5.342 (0.563, 10.120) *
Q3	2.743 (0.248, 5.238) *	7.282 (3.965, 10.599) **	8.935 (3.873, 13.997) *
Q4	9.116 (6.658, 11.574) **	11.326 (7.920, 14.732) **	12.376 (6.764, 17.988) **
*p*-value for trend	<0.001	<0.001	<0.001
DBP			
Continuous	0.115 (0.038, 0.193) *	0.132 (0.041, 0.223) *	0.184 (0.018, 0.349) *
Q1	Reference	Reference	Reference
Q2	0.254 (−1.436, 1.944)	−1.279 (−3.368, 0.810)	1.147 (−2.203, 4.498)
Q3	0.168 (−1.551, 1.887)	0.346 (−1.773, 2.466)	2.216 (−1.333, 5.765)
Q4	2.83 (1.137, 4.523) *	3.007 (0.831, 5.182) *	2.727 (−1.208, 6.661)
*p*-value for trend	0.004	0.005	0.031

* *p* < 0.05; ** *p* < 0.001 Model was adjusted for age, gender, vegetable consumption, fruit consumption, smoking habits, alcohol drinking, sleeping duration, and mid–high sports hours.

## Data Availability

The data supporting the conclusions of this article will be made available by the authors, without undue reservation.

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
