# Peer review of "Association between Body Fat and Elevated Blood Pressure among Children and Adolescents Aged 7–17 Years: Using Dual-Energy X-ray Absorptiometry (DEXA) and Bioelectrical Impedance Analysis (BIA) from a Cross-Sectional Study in China"

_ijerph, 2021, doi:10.3390/ijerph18179254_

Round 1
Reviewer 1 Report
Congratulations, a very good study. Please make the suggested corrections in your document. Also review your statements in the introduction.
I found the manuscript interesting and very well done. However, I suggest the following minor changes and corrections:
- Introduction—Line 36-the term ”….cardiovascular diseases”. Cardiovascular disease can be referred to a number of conditions, heart disease, heat attack, etc. The authors should, instead, emphasize more the pathological mechanisms involved in hypertension.
- Study population:
The grouping of children and adolescents ranging from 7 to 17 years old, presents too large age difference to be grouped as one study group. It would be best to divide in smaller age groups around the mean age, such as perhaps, two or three different groups of both, boys and girls; unless the authors did not see any significant differences in the test results among the children ranging from 7 to 17 years old. I would suggest to present the table 1 showing the test results for the different age groupings.
- Definitions of EBP
Line 127… According to the national standards of EBP threshold in China. It would be helpful to know the Chinese national standard of EBP threshold.
- Results
- Line 170—a p-value of 0.269 does show a statistically significant differences for DBP (sitting position). This p-value is definitely greater than 0.05
- Table 1. The values shown for BP, EBP. SBP. ESBP, DBP and EDBP, in mmHg, do not make sense, the values are too high.
Author Response
1. I found the manuscript interesting and very well done. However, I suggest the following minor changes and corrections:
Thank you for your comments, we appreciate your thoughtful and useful comments. Replies to the suggestions are as follows.
2. Introduction—Line 36-the term ”….cardiovascular diseases”. Cardiovascular disease can be referred to a number of conditions, heart disease, heat attack, etc. The authors should, instead, emphasize more the pathological mechanisms involved in hypertension.
Response: Thank you for your comments and suggestions, and we agree with your suggestions.
We have revised it in the article as follows: “Hypertension had caused serious disease burdens in China, and was associated with adverse cardiac changes, adipose tissue factor changes and vascular damage [4]. The increase in BP was affected by multiple factors, such as excessive intake of salt, reduced physical activity, and excessive psychological stress, which could further account for the development of hypertension[5].”
[4] KHOURY M, URBINA EM. Hypertension in adolescents: diagnosis, treatment, and implications. Lancet Child Adolesc Health. 2021, 5(5):357-66.
[5] DESAI A N. High Blood Pressure. JAMA 2020, 324(12): 1254-5.
3. Study population:
The grouping of children and adolescents ranging from 7 to 17 years old, presents too large age difference to be grouped as one study group. It would be best to divide in smaller age groups around the mean age, such as perhaps, two or three different groups of both, boys and girls; unless the authors did not see any significant differences in the test results among the children ranging from 7 to 17 years old. I would suggest to present the table 1 showing the test results for the different age groupings.
Response: Thank you for pointing out this point. We have added results of age groups (7-12years, 13-15years, and 16-17years) with chi-square test results in Table 1. These three age groups represented the primary school stage, middle school stage and middle high school stage.
4. Definitions of EBP
Line 127… According to the national standards of EBP threshold in China. It would be helpful to know the Chinese national standard of EBP threshold.
Response: Thank you for pointing out this point. We do agree with you and we had revised the description of the Chinese national standard of EBP threshold: “The average of SBP and DBP values were calculated by two measurements. According to the national reference of EBP in China[17], ESBP (elevated systolic blood pressure) and EDBP (elevated diastolic blood pressure) were defined as average measured SBP and DBP greater than or equal to the 90th percentile (on the basis of age, sex, and height percentiles), respectively. EBP was defined as ESBP or EDBP of children and adolescents.”
5. Results: Line 170-a p-value of 0.269 does show a statistically significant differences for DBP (sitting position). This p-value is definitely greater than 0.05
Table 1. The values shown for BP, EBP. SBP. ESBP, DBP and EDBP, in mmHg, do not make sense, the values are too high.
Response:We truly appreciate your comments, we re-examined the results and did not find errors.
“Boys tended to have higher SBP (sitting position) compared with girls (p = 0.001), while no significant differences were observed for DBP (sitting position) between boys and grils (p = 0.269).”
In addition, we apologize for causing such confusion, the values shown for BP, EBP, SBP, ESBP, DBP and EDBP in Table 1, was presented the form of n (%). We added the detailed notes in the Table 1.
Reviewer 2 Report
The Authors may included the children's from other parts of the world for better results.
Author Response
Response: Thank you very much for pointing out this point. We do agree with you and we also considered this limitation in the article. As we mentioned in the line 341-342 as below, the sample size was not large enough which limited the extrapolation of the conclusion. As you suggested, if we can include the children from other parts of the world, the results would be better and reliable. We should enlarge the sample size to provide research ideas for the future identification and intervention of children and adolescents between high blood pressure and high body fat percentage next time.
Reviewer 3 Report
Please see the attached files for my comments.

Author Response
Thank you very much for your comments and suggestions. The reply to the review report in the attachment.

Round 2
Reviewer 3 Report
Please see the attached comments. The manuscript has been improved extensively. Please fix few issues mentioned in the document attached and it should be sufficient for me.
I don't need to see this again. Congratulations to the authors for such a wonderful study.

Author Response
1. why 7-12, 13-15, 16-17, why not 7-9, 10-14, 15-17? provide rationale. add a copy of this survey in the supplementary file. This way in future, if someone wants to replicate findings, they can use it.
Response: Thank you for pointing out this point. Both those two methods of age stratification were reasonable. This method (7-12years, 13-15years and 16-17years) was divided according to the students’ grades (primary school stage (grade 1 to 6), middle school stage (grade 1 to 3), middle high school stage (grade 1 to 3)), which conducted to health education in schools. The other method (7-9years, 10-14years, 15-17years) was divided according to similar age groups, which considered the similar growth and development. In this article, we adopted the method as follows.
We divided the age into three groups (7-12years, 13-15years and 16-17years). Generally, these three age groups represented the primary school stage (grade 1 to 6), middle school stage (grade 1 to 3), middle high school stage (grade 1 to 3) in China.
2. the protocol for these plots should be briefly described in methods
Response: Thank you for your comments. We have added the details in Statistical Analysis.
Bland-Altman plots were created using mean and the difference to predict the limits of agreement of body fat mass and BF% between DEXA and dual frequency BIA machine measurements.